# Association between Obstructive Sleep Apnea and SYNTAX Score

**DOI:** 10.3390/jcm9103314

**Published:** 2020-10-15

**Authors:** Sayaki Ishiwata, Yasuhiro Tomita, Sugao Ishiwata, Koji Narui, Hiroyuki Daida, Takatoshi Kasai

**Affiliations:** 1Department of Cardiovascular Medicine, Juntendo University School of Medicine, Tokyo 113-8421, Japan; s-ishiwata@juntendo.ac.jp (S.I.); daida@juntendo.ac.jp (H.D.); kasai-t@mx6.nisiq.net (T.K.); 2Cardiovascular Respiratory Sleep Medicine, Juntendo University Graduate School of Medicine, Tokyo 113-8421, Japan; 3Department of Cardiovascular Medicine, Toranomon Hospital, Tokyo 105-8470, Japan; sishii@toranomon.gr.jp; 4Department of Sleep Medicine, Toranomon Hospital, Tokyo 105-8470, Japan; sas6344@nifty.com; 5Sleep and Sleep-Disordered Breathing Center, Juntendo University Hospital, Tokyo 113-8421, Japan

**Keywords:** coronary artery disease, obstructive sleep apnea, severity

## Abstract

Obstructive sleep apnea (OSA) is related to an increased risk of cardiovascular diseases, including coronary artery disease (CAD). We investigated the association between OSA and the severity of CAD by assessing coronary angiography findings. We retrospectively analyzed patients who underwent their first coronary angiography to evaluate CAD and polysomnography (PSG) to investigate the severity of OSA in our hospital from March 2002 to May 2015. The severity of CAD was determined based on coronary angiography findings using the SYNTAX score. The patients were divided into two groups according to the apnea-hypopnea index (AHI): mild OSA (AHI < 15/h) and moderate-to-severe OSA (AHI ≥ 15/h). Overall, 98 patients were enrolled. The SYNTAX score was significantly different between the two groups (*p* = 0.001). After adjustment for other risk factors, including age, sex, obesity, hypertension, hyperlipidemia, diabetes mellitus, smoking status, and family history of CAD, moderate-to-severe OSA significantly correlated to the SYNTAX score (partial correlations = 0.24, *p* = 0.039). These results suggest that the severity of CAD is related to moderate-to-severe OSA.

## 1. Introduction

Identifying high-risk patients with coronary artery disease (CAD) and associated risk factors is crucial in the planning of treatment. Obstructive sleep apnea (OSA) is a common disease, and CAD is an important comorbidity of OSA. The prevalence of OSA is reported to be 9–14% in men and 4–7% in women in the general population [1,2]. In contrast, approximately 24% of patients with CAD reportedly have OSA based on a cutoff apnea-hypopnea index (AHI) > 15 [3]. Continuous positive airway pressure is the most effective treatment for reducing the number of respiratory events and improving OSA-related symptoms; furthermore, it can reduce the risk of cardiovascular disease [4]. There is some mechanism underlying the relationship between OSA and cardiovascular diseases. In patients with OSA, the pathophysiological consequences of OSA, such as the intrathoracic negative pressure generated by upper airway obstruction while patients are asleep and overactivation of the sympathetic nervous system and oxidative stress, increase the risk of hypertension, CADs, heart failure, arrhythmia, stroke [5], and other cardiovascular risk factors such as diabetes mellitus and hypercholesterolemia [6,7].

One study in which the relationship between the severity of OSA and CAD were assessed in patients hospitalized for acute coronary syndrome (ACS) and who underwent polysomnography revealed that the severity of OSA was associated with that of CAD as evaluated based on the SYNTAX score [8]. The SYNTAX score [9], a scoring system to objectively assess the complexity of the coronary artery based on coronary angiography (CAG) findings using a calculator system, has been used in recent years. The SYNTAX score is an angiographic score that predicts CAD severity and is used to choose between percutaneous coronary intervention (PCI) and coronary artery bypass grafting for revascularization. The severity of CAD as assessed by the number of obstructed coronary vessels based on CAG or coronary computed tomography (CCT) reportedly contributes to worse long-term outcome in patients with both ACS and stable CAD [10,11]. However, the relationship between the severity of OSA and stable CAD has not been elucidated. This study aimed to investigate the association between OSA and the severity of CAD and identify factors predictive of the severity of CAD.

## 2. Experimental Section

### 2.1. Subjects

Data of subjects who underwent in-laboratory diagnostic polysomnography because of clinical indications and who underwent their first CAG for the evaluation of CAD in our hospital between March 2002 and May 2015 were screened. We set the time interval between CAG and polysomnography to be within 1 year. Among them, data of subjects who met the inclusion criteria were analyzed. The inclusion criteria were as follows: (1) men or women aged ≥ 20 years and (2) the interval between CAG and polysomnography is within 1 year. Exclusion criteria were as follows: (1) diagnosed predominant central sleep apnea (CSA); (2) diagnosed with and/or treated for neuromuscular disease; (3) undergoing dialysis; (4) diagnosed with heart failure; or (5) treated for OSA before polysomnography. The protocol of this study was approved by the Ethics Committee of Toranomon Hospital, and this study complied with the Declaration of Helsinki.

### 2.2. Polysomnography

Polysomnography (SomnoStar Alpha Sleep System, SensorMedics Corp., Yorba Linda, CA, USA) was performed for patients with CAD. We defined OSA as an AHI > 15. All patients underwent overnight polysomnography using a digital polygraph system during hospitalization for the evaluation of OSA. Definitions and scoring methods were based on the American Academy of Sleep Medicine manual version 2.2 [12]. We monitored thoracoabdominal motion using respiratory inductance plethysmography. Air flow was measured using an oronasal thermal airflow sensor and nasal pressure cannula. Oxyhemoglobin saturation was monitored using an oximeter. PSG was performed and scored by sleep physician who was blinded to the patients’ coronary angiogram and blood test results.

The severity of OSA was assessed based on the frequency of apneas and hypopneas per hour of sleep (AHI). The subjects were divided into those with < 15 events of apnea and hypopnea per hour of sleep (no-to-mild OSA group) and those with ≥ 15 AHI events (moderate-to-severe OSA group). The time spent with oxygen saturation < 90% (T90) was defined as the percentage of total sleep time.

Predominant CSA was defined as follows: central AHI (CAHI) ≥ obstructive AHI + mixed AHI and CAHI ≥ 15. Obstructive AH (apnea-hypopnea) was defined as apnea and hypopnea caused by the obstruction of the upper airway. Mixed AH was defined as obstructive AH occurring after central AH.

### 2.3. CAG and SYNTAX Score

The severity of CAD was determined based on CAG findings using the SYNTAX score, a scoring system of CAD severity that accounts for the complexity of lesion morphology. The SYNTAX score is one of the most commonly used indicators for the risk stratification of CAD patients. The SYNTAX score describes coronary vasculature complexity based on lesion location and characteristics, including total occlusion, trifurcation, bifurcation, severe tortuosity, length >20 mm, heavy calcification, and presence of thrombus. The risk of CAD was defined as follows: a SYNTAX score ≤22 is classified as low risk, that between 23 and 32 as intermediate risk; and that ≥33 as high risk. SYNTAX scores were determined based on the decision by two cardiologists.

### 2.4. Other Data Collection

We collected data related to risk factors (obesity, hypertension, hyperlipidemia, diabetes mellitus) and patient characteristics (age, sex, smoking history, family history of CAD) by reviewing medical charts. The risk factors were as follows: (1) obesity as defined by a body mass index (BMI) >25 kg/m^2^, (2) hypertension defined as systolic blood pressure >140 mmHg and/or diastolic blood pressure >90 mmHg, (3) hyperlipidemia defined as low density lipoprotein level >140 mg/dL, and (4) diabetes mellitus defined as HbA1c > 6.5% and/or taking antidiabetic agents.

### 2.5. Statistical Analyses

Continuous valuables are expressed as mean ± standard deviation or median with interquartile range. Categorical valuables are presented as numbers and percentages. When comparing patient’s characteristics between two groups, the χ^2^ test or Fisher’s exact test was used for categorical variables and the t-test or Mann-Whitney U-test was used for continuous variables. Since SYNTAX scores had a skewed distribution, log-transformed variables adding 0.1 to each SYNTAX score were used in the regression analyses. Relationships between SYNTAX score and characteristics, including age, sex, obesity, hypertension, hyperlipidemia, diabetes mellitus, smoking status, and family history of CAD, in addition to OSA severity (no-to-mild OSA or moderate-to-severe OSA), were assessed using univariable regression analyses. Multivariable stepwise regression analyses (*p* < 0.05 to enter and *p* > 0.1 to remove) were performed with the SYNTAX score as a dependent variable and variables with *p* values < 0.05 in the univariable analyses. A *p* value < 0.05 was considered statistically significant. All statistical analyses were performed using R software version 3.4.3 (R Core Team, Vienna, Austria). Because the sequence of coronary angiography and polysomnography may affect the relationship between OSA and SYNTAX score, the interaction between the sequence and the relationship between moderate to severe OSA and SYNTAX score was assessed.

## 3. Results

In total, 242 patients underwent CAG and PSG from March 2002 to May 2015. Patients with more than 1 year between CAG and PSG were excluded. Finally, 98 patients were analyzed. The mean age of the included subjects was 58.3 years, and 95 (92%) patients were male; 75 patients (77%) had moderate-to-severe OSA. The characteristics of the subjects are shown in Table 1. The SYNTAX score was significantly greater in patients with moderate-to-severe OSA than in those with no-to-mild OSA (median and interquartile range (IQR), 4.0 and 13.5 vs. 0 and 1, respectively, *p* = 0.001) (Figure 1). T90 was significantly greater in patients with moderate-to-severe OSA than in those with no-to-mild OSA (median and interquartile range (IQR), 10.9 and 23.2 vs. 0.3 and 1.25, respectively, *p* = 0.001). Arousal was significantly greater in patients with moderate-to-severe OSA than in those with no-to-mild OSA (mean ± SD, 25.3 ± 8.1 vs. 43.5 ±19.7, respectively, *p* = 0.001). Ten (10%) patients underwent CAG because of suspected ACS in the moderate-to-severe OSA group and none in the no-to-mild OSA group. There was no significant differences with respect to age, sex, BMI, history of diabetes mellitus, hyperlipidemia, hypertension, current smoking, family history of ischemic heart disease, obesity, total sleep time (TST), rapid eye movement (REM) or stage N3 between the no-to-mild and moderate-to-severe OSA groups. Ten variables (i.e., age, sex, BMI, history of diabetes mellitus, hyperlipidemia, hypertension, current smoking, family history of ischemic heart disease, moderate-to-severe OSA, and SYNTAX score) were included the multivariable analysis. The results of the multivariable analysis are shown in Table 2. In the multivariable linear regression model, older age, presence of hyperlipidemia, and presence of moderate-to-severe OSA were significantly independently correlated to the SYNTAX score (partial correlations = 0.24, *p* = 0.039). There was no interaction between the sequence (coronary angiogram first or polysomnography first) and the relationship between moderate to severe OSA and SYNTAX score (*p* = 0.596).

## 4. Discussion

The study findings provide new insight into the relationship between the severity of OSA and that of CAD in patients who underwent PSG owing to a clinical indication and who underwent their first CAG for the evaluation of CAD. SYNTAX scores were significantly higher in patients with moderate-to-severe OSA than in those with no-to-mild OSA. Second, moderate-to-severe OSA was associated with CAD severity as assessed using the SYNTAX score in the multivariable linear regression analysis. This study raises the possibility that CAD in patients with OSA is anatomically complex. Patients with severe CAD may have undiagnosed OSA; thus, a screening approach to identify OSA is important for these patients.

To our knowledge, this is the first study to investigate whether moderate-to-severe OSA is correlated to CAD severity stratified based on the SYNTAX score in patients with stable CAD. In general, patients with OSA have more severe atherosclerosis than those without OSA [13]. Approximately 38–65% of patients with CAD have OSA [14] and sleep disordered breathing (SDB) is better correlated with CAD in men than in women [15].

The coronary artery vasculature can be evaluated using either non-invasive [16] or invasive procedures, such as CAG [17]. Several studies have investigated cardiovascular atherosclerosis using non-invasive methods, such as carotid artery intima-media thickness (carotid IMT) measured using ultrasonography and CCT. Carotid IMT allows the estimation of early atherosclerosis of the coronary artery and has been a useful indicator of prognosis in patients with CAD [18]. A Japanese study of OSA patients using PSG reported that carotid IMT was significantly higher in the moderate-to-severe OSA group than in the mild OSA groups (1.16 ± 0.07 mm vs. 0.92 ± 0.07 mm, respectively; *p* < 0.003) [19]. CCT is a non-invasive procedure for assessing the vascular morphology of the coronary artery and plaque size and has predictive utility for patients with CAD. A study of 202 patients with several coronary risk factors who underwent overnight PSG owing to clinical indications and CCT for the evaluation of coronary artery calcification (CAC) found that the CAC score was significantly higher in the highest AHI quartile than in the lower quartile groups. In the multivariable analysis, severe OSA was independently correlated to the CAC score [20]. CCT can also detect non-calcified plaques. In a healthy population without overt coronary risk factors who underwent PSG because of indications for SDB, moderate-to-severe OSA correlated significantly more with the increasing non-calcified plaque volume on CCT than mild OSA [21]. Our study evaluated sleep-disordered breathing using PSG, which is the most precise diagnostic tool for assessing OSA.

A large prospective study of patients presenting with CAD performed CAG and intravascular ultrasound and found that the independent determinant of total atheroma volume of the target lesion in the coronary artery was moderate-to-severe OSA [22].

The severity of CAD has been conventionally assessed using the number of diseased vessels and proportion of coronary artery stenosis in the left main trunk lesion based on CAG findings [23,24]. In a systematic review, OSA was reported to be a risk factor for patients with CAD and could contribute to the progression of CAD [25].

CAG was established as a procedure to evaluate the coronary artery in patients at a high risk of CAD. In one study from Japan, the association between SDB defined as ≥5 events of 3% oxygen desaturation (ODI3%) per hour and coronary atherosclerosis as measured using the Gensini score was evaluated. The Gensini score is calculated as the sum of the score for each lesion. The points are obtained from the stenosis of the lumen and multiplied by factors associated with the importance of lesion position in the coronary artery [26]. The Gensini score was associated with an increase in ODI (R = 0.45, *p* = 0.01), and the Gensini score was higher in the group with ≥15 ODI events per hour than in that with <5 ODI events per hour [27]. In another study involving patients with a clinical indication for CAG because of suspected CAD who underwent portable polygraphy, more severe CAD as determined based on a higher Gensini score was associated with an AHI of ≥15 events per hour, but a dose-response relationship was not proven [28]. Subsequently, in a study of CAD patients who underwent PCI, the SYNTAX score was presented as a useful risk evaluation index [29,30].

In the present study, we enrolled 10 patients (10%) who underwent emergency CAG for ACS. In one study evaluating the relationship between moderate-to-severe OSA and the severity of CAD as assessed using the SYNTAX score in ACS patients, moderate-to-severe OSA was associated with the severity of CAD [8]. However, another study investigating whether OSA severity as assessed in a screening sleep test is related to the severity of CAD as assessed using the SYNTAX score in acute myocardial infarction patients found no significant association between AHI and angiographical coronary artery severity [31]. In the present study, patients with stable CAD account for a majority of participants, but we included small numbers of patients with ACS (10%). The effect of ACS on SDB status was assessed when PSG was performed in acute phase of ACS, and worsening of SDB and generation of CSA were noted. However, PSG was not performed in the acute phase of CAD, although in the stable phase, we could exclude the effect of ACS on SDB.

### Limitation

This study has some limitations. First, this was a single-center retrospective study of a small cohort and this could have affected our findings. Moreover, we could not determine a causal relationship between the severity of CAD and OSA.

Second, we should consider the influence of gender disproportion because over 90% of the patients were male. However, taking into account the gender differences in prevalence of OSA and CAD, the percentage does not seem to be too large.

Third, there were several unmeasured factors, such as echocardiography findings, including systolic and diastolic function, New York Heart Association functional classification, and medication use (e.g., angiotensin-converting enzyme inhibitors, angiotensin II receptor blockers, aldosterone blockers, beta-blockers, and diuretics). Moreover, we could not obtain data regarding the degree of sleepiness as assessed using the Epworth Sleepiness Scale or CAD evaluation using the SYNTAX II score, which includes more information regarding patient background. To assess the long-term prognosis of CAD patients, additional factors associated with patient background are required [32,33,34].

Fourth, we did not evaluate the sequential effect of OSA severity on CAD as assessed using the SYNTAX score or the effect of positive airway pressure treatment on CAD. The significance of interventions for OSA in these patients has not yet been elucidated. There are some pathophysiological mechanisms involved in this effect, including changes in intrathoracic pressure and intermittent hypoxia-generating oxidative stress, overactivation of the sympathetic nervous system, and inflammation related to transcription factors [13].

Fifth, some of the patients who underwent PSG before CAG could be benefit from the treatment for OSA. However, PSG was performed after coronary angiography in 77 patients (79%), and the same results were observed in this subgroup. Thus, the confounding influence of the test sequence was considered as little. Thus, a further study in which the relationship between the severity of OSA and CAD is prospectively investigated is required.

## 5. Conclusions

The SYNTAX score was significantly higher in patients with moderate-to-severe OSA than in those with no-to-mild OSA. Moderate-to-severe OSA was associated with the SYNTAX score in a multivariable linear regression analysis. These results suggest that more severe CAD is related to moderate-to-severe OSA.

## Figures and Tables

**Figure 1 jcm-09-03314-f001:**
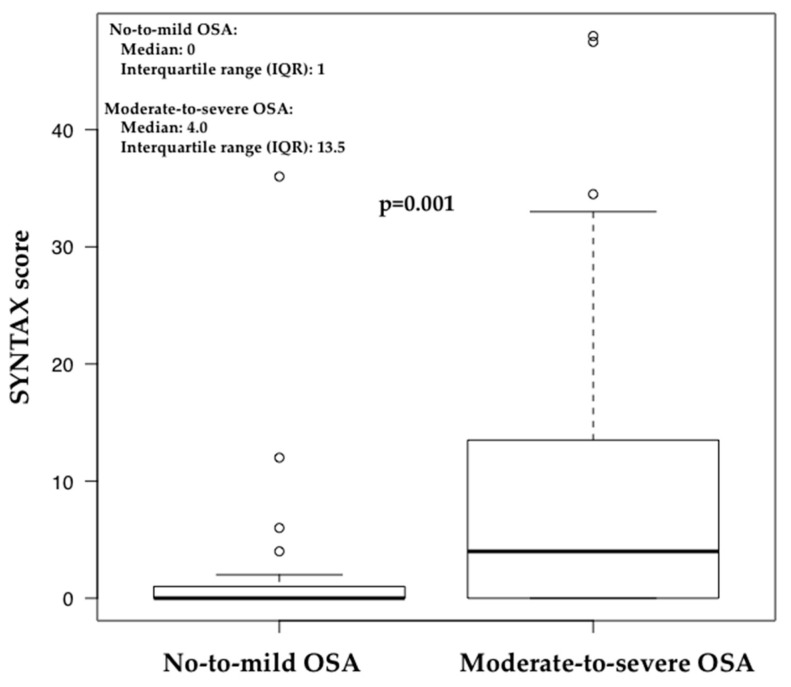
Box and whisker plot demonstrating the distribution of SYNTAX scores according to the severity of SA. The SYNTAX score was significantly greater in patients with moderate-to-severe SA than in those with no-to-mild SA (median and interquartile range (IQR), 4.0 and 13.5 vs. 0 and 1, respectively, *p* = 0.001).

**Table 1 jcm-09-03314-t001:** Baseline characteristics of all subjects.

	No-to-Mild OSA (*n* = 23)	Moderate-to-Severe OSA (*n* = 75)	*p* Value
Age, years	56 ± 14	59 ± 12	0.232
Male sex, *n* (%)	22 (96%)	68 (91%)	0.672
BMI, kg/m^2^	25.1 ± 2.9	26.4 ± 5.1	0.317
Diabetes mellitus, *n* (%)	5 (25%)	26 (39.4%)	0.295
Family history, *n* (%)	5 (25%)	8 (12.3%)	0.175
Hyperlipidemia, *n* (%)	9 (45%)	37 (55.2%)	0.454
Hypertension, *n* (%)	12 (52.2%)	35 (47.3%)	0.812
Current smoking, *n* (%)	6 (26.1%)	17 (23.3%)	0.784
Obesity, *n* (%)	9 (39.1%)	43 (57.3%)	0.155
TST, min	356.6 ± 56.6	309.8 ± 104.5	0.063
REM, %TST	13.9 ± 7.1	10.9 ± 6.0	0.068
Stage N3, %TST	9.9 ± 7.3	9.3 ± 9.5	0.796
T90, %	0.3 (1.25)	10.9 (23.2)	<0.001
Arousal index,/h	25.3 ± 8.1	43.5 ± 19.7	<0.001
AHI,/h	8.7 ± 4.0	41.0 ± 20.0	<0.001
OAHI,/h	7.3 ± 4.3	36.2 ± 19.2	<0.001

Variables are expressed as mean ± standard deviation or *n* (%). AHI: apnea hypopnea index, BMI: body mass index, OAHI: obstructive apnea hypopnea index, OSA: obstructive sleep apnea, REM: rapid eye movement, TST: total sleep time, T90: time spent with oxygen saturation < 90%.

**Table 2 jcm-09-03314-t002:** Results of multivariable linear regression analysis.

	Partial Correlations	*p*-Value
Age, years	0.29	0.011
Male sex	0.09	0.423
Obesity (BMI ≥ 25 kg/m^2^)	0.13	0.245
Hypertension	0.08	0.474
Hyperlipidemia	0.32	0.005
Diabetes mellitus	0.19	0.103
Current smoking	0.07	0.568
Family history of CAD	0.10	0.387
Moderate-to-severe OSA	0.24	0.039

BMI: body mass index, CAD: coronary artery disease, OSA: obstructive sleep apnea.

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
