# Peer review of "Association between Obstructive Sleep Apnea and SYNTAX Score"

_jcm, 2020, doi:10.3390/jcm9103314_

Round 1
Reviewer 1 Report
In this interesting retrospective cohort study, Ishiwata et al explore the relationship between severity of sleep apnea and complexity and severity of coronary atherosclerosis as measured by the SYNTAX score. All patients who underwent polysomnography and a coronary angiogram within 12 months of each other at a single center between 03/2002 to 05/2015 were included barring the presence of exclusion criteria which are reasonable.
98 patients overall were included in the analysis and the authors note a statistically significant relationship between severity of sleep apnea as measured by apnea-hypopnea index score and SYNTAX score.
Background information given is sound and the study was appropriately registered with their institution after ethical approval had been sought. Discussion is sound, with comparisons made with other related studies which are appropriately referenced. Overall the manuscript makes a good case for the relationship between sleep apnea severity and SYNTAX score.
Issues that need to be considered & addressed:
- Line 49 – describes the SYNTAX score as ‘new’ despite it being used for around 10 years and that the authors also describe non-use of the updated SYNTAX 2 score as a limitation
- The methodology section needs work and further detail. There is no information given as to who was interpreting the polysomnography assessments nor the coronary angiogram to calculate SYNTAX scores which are known to produce quite variable results. There is no description of any (if it was done) blinding of assessors to sleep apnea status or vice versa and no mention of assessment of inter or intra-observer agreement during analysis which is likely the biggest confounder at play. In the reviewer’s opinion, this section requires significant expansion.
- The authors do describe results of PSG in terms of AHI score however do not mention nor describe any of the other multitude of variables gleaned from the PSG which may be of interest
- The authors mention not assessing the effect of treatment of OSA nor do they expand on how many of the patients (if any) were on treatment for OSA following their PSG but before the time of their angiography which again may introduce confounders. One option would potentially be to perform a sub analysis of those patients who underwent their angiogram first and had their PSG at a later date.
- A note is made of the fact that over 90% of patients were male. This should likely be added to limitations.
- Line 156 – the abbreviation SDB needs definition
- Line 156 – grammatical error – the word ‘to’ ought to be removed “correlated to with…”
Overall this is an interesting study which appears to have well performed but requires some of the above to be considered and adjusted prior to publication.
Author Response
Dear editor:
Thank you for your consideration. We have read the reviewers’ comment and corrected accordingly.
Dear Reviewer 1:
Thank you for your suggestion. We have modified the manuscript according to your comments.
- Line 49 – describes the SYNTAX score as ‘new’ despite it being used for around 10 years and that the authors also describe non-use of the updated SYNTAX 2 score as a limitation
-> We agree with you and removed ‘new’ in Line 49.
- The methodology section needs work and further detail. There is no information given as to who was interpreting the polysomnography assessments nor the coronary angiogram to calculate SYNTAX scores which are known to produce quite variable results. There is no description of any (if it was done) blinding of assessors to sleep apnea status or vice versa and no mention of assessment of inter or intra-observer agreement during analysis which is likely the biggest confounder at play. In the reviewer’s opinion, this section requires significant expansion.
-> PSG was performed and scored by sleep technologists who were blinded to the patients’ coronary angiogram and blood test results. A sleep physician who was also blinded to the patients’ coronary angiogram and blood test results interpreted findings of PSG. SYNTAX scores were determined based on the decision by 2 cardiologists who were blinded to the results of PSG.
- The authors do describe results of PSG in terms of AHI score however do not mention nor describe any of the other multitude of variables gleaned from the PSG which may be of interest
-> We have added several EEG parameters as well as % of time spent SO2<90% (T90) per total sleep time as an index of oxygenation in the baseline characteristics. T90 was significantly greater in patients with moderate-to-severe OSA than in those with no-to-mild OSA (median and interquartile range [IQR], 10.9% and 23.2% vs. 0.3% and 1.25%, respectively, P = 0.001).
The authors mention not assessing the effect of treatment of OSA nor do they expand on how many of the patients (if any) were on treatment for OSA following their PSG but before the time of their angiography which again may introduce confounders. One option would potentially be to perform a sub analysis of those patients who underwent their angiogram first and had their PSG at a later date.
->Because the sequence of coronary angiography and polysomnography may affect the relationship between OSA and SYNTAX score, the interaction between the sequence and the relationship between moderate to severe OSA and SYNTAX score was assessed. PSG was performed after coronary angiography in 77 patients (79%). There was no interaction between the sequence (coronary angiography first or polysomnography first) and the relationship between moderate to severe OSA and SYNTAX score (p=0.596). We did not present the results of 77 patients in manuscript because the same results were observed in this subgroup.
Table 1. Baselinecharacteristicsof all subjects.(n=77)
|
|
No-to-mild OSA (n=16) |
Moderate-to-severe OSA (n=61) |
P value |
|
Age, years |
54±14 |
59±12 |
0.119 |
|
Male sex, n (%) |
14 (87.5 %) |
56 (91.8 %) |
0.318 |
|
BMI, kg/m2 |
25.4±3.1 |
25.9±4.8 |
0.707 |
|
Diabetes mellitus, n (%) |
4 (25 %) |
17 (27.9 %) |
0.301 |
|
Family history, n (%) |
3 (18.6 %) |
5 (8.2 %) |
0.754 |
|
Hyperlipidemia, n (%) |
6 (37.5 %) |
28 (45.9 %) |
0.313 |
|
Hypertension, n (%) |
6 (37.5 %) |
31 (50.8 %) |
0.907 |
|
Current smoking, n (%) |
4 (25 %) |
16 (26.2 %) |
0.646 |
|
Obesity, n (%) |
7 (43.8%) |
32 (52.5 %) |
0.866 |
|
T90, % |
0.2 [1.9] |
8.4 [18.7] |
<0.001 |
|
AHI, /h |
8.6±4.1 |
41.2±18.5 |
<0.001 |
|
OAHI, /h |
7.4±4.2 |
35.2±18.4 |
<0.001 |
Table 2.Resultsof multivariable linear regression analysis(n=77)
|
|
Partial correlations |
P value |
|
Age, years |
0.28 |
0.037 |
|
Male sex |
0.09 |
0.530 |
|
Obesity (BMI ≥25 kg/m2) |
0.11 |
0.425 |
|
Hypertension |
0.12 |
0.383 |
|
Hyperlipidemia |
0.33 |
0.014 |
|
Diabetes mellitus |
0.09 |
0.516 |
|
Current smoking |
0.10 |
0.461 |
|
Family history of CAD |
0.07 |
0.637 |
|
Moderate-to-severe OSA |
0.18 |
0.199 |
- A note is made of the fact that over 90% of patients were male. This should likely be added to limitations.
-> We have added this fact into Limitation.
- Line 156 – the abbreviation SDB needs definition
-> We have added ‘sleep disordered breathing (SDB) ’.
- Line 156 – grammatical error – the word ‘to’ ought to be removed “correlated towith…”
-> We have corrected as you suggested.
Reviewer 2 Report
This is concise, properly constructed study worth of publication. Small editorial mistakes and description of methodology need slight improvement.
- "These reults suggest that...OSA is related to severity of CAD" - in fact the meaning of reults is just opposite: the severity/score/intensity of CAD is related to OSA, not vice versa
- A little bit more attention should be paid to the abbreviations used in the text, e.g.: instead of OSA at the end appears SA; first it was polysomnography (without abbreviation) and next PSG
- A reader cannot be sure how were the patients recruited: because they were seeking help for sleep breathing disorders of cardiac problems or both?
- A reader cannot be sure whether polysomnography was used (as suggested in the lines 81 or 173-174 or respiratory polygraphy (as in the description of the nocturnal study).
- It is not clear why arterial oxygen saturation was not taken into consideration, although it was measured. This would be important information. Could the Authors add this data to RESULTS or list the problem among the limitations of the study (and explaining why SaO2 was not revealed)?
- line 156: small grammatical error
- Unnecessarily the trade name of PSG device was used twice
- line 84-85: incorrectly defined central sleep apnea
Author Response
Dear editor:
Thank you for your consideration. We have read the reviewers’ comment and corrected accordingly.
Dear Reviewer 2:
Thank you for your suggestion. We have modified the manuscript according to your comments.
- "These reults suggest that...OSA is related to severity of CAD" - in fact the meaning of reults is just opposite: the severity/score/intensity of CAD is related to OSA, not vice versa
-> We have corrected as you suggested.
- A little bit more attention should be paid to the abbreviations used in the text, e.g.: instead of OSA at the end appears SA; first it was polysomnography (without abbreviation) and next PSG
-> We have corrected as you suggested.
- A reader cannot be sure how were the patients recruited: because they were seeking help for sleep breathing disorders of cardiac problems or both?
-> Both patients were included. We presented this explanation into 2.1. Subjects section.
- A reader cannot be sure whether polysomnography was used (as suggested in the lines 81 or 173-174 or respiratory polygraphy (as in the description of the nocturnal study).
-> In our study, overnight full-PSG was performed and presented in 2.2. Polysomnography section.
- It is not clear why arterial oxygen saturation was not taken into consideration, although it was measured. This would be important information. Could the Authors add this data to RESULTS or list the problem among the limitations of the study (and explaining why SaO2 was not revealed)?
-> We have included percent of time spent SO2<90% per total sleep time (T90) as an index of oxygen desaturation in the baseline characteristics. The T90 was significantly greater in patients with moderate-to-severe OSA than in those with no-to-mild OSA (median and interquartile range [IQR], 10.9% and 23.2% vs. 0.3% and 1.25%, respectively, P = 0.001). In univariable analysis, T90 was not significant correlate to the SYNTAX score (p=0.244).
- line 156: small grammatical error
-> We have corrected ‘correlated to with’ to ‘correlated with’.
- Unnecessarily the trade name of PSG device was used twice
->We have removed one of them in 2.2. Polysomnography section.
- line 84-85: incorrectly defined central sleep apnea
-> We included patients with an CAHI of less than 15 events per hour of sleep unlike the usual definition because we would like to exclude only the typical CSA.